# In Vitro and In Vivo Testing of Decellularized Lung and Pancreas Matrices as Potential Islet Platforms

**DOI:** 10.3390/ijms26146692

**Published:** 2025-07-12

**Authors:** Alexandra Bogomolova, Polina Ermakova, Arseniy Potapov, Artem Mozherov, Julia Tselousova, Ekaterina Vasilchikova, Alexandra Kashina, Elena Zagaynova

**Affiliations:** 1Lopukhin Federal Research and Clinical Center of Physical-Chemical Medicine, Moscow 119334, Russia; baleksandra@icloud.com (A.B.); ezagaynova@gmail.com (E.Z.); 2Federal State Budgetary Institution of Higher Education, “Privolzhsky Research Medical University” of the Ministry of Health of Russia, Nizhny Novgorod 603082, Russia; bardina_p@pimunn.net (P.E.); potapov_al@pimunn.net (A.P.); julia@yandex.ru (J.T.); vasilchikova_ea@pimunn.net (E.V.); 3Institute for Regenerative Medicine, Sechenov First Moscow State Medical University (Sechenov University), Moscow 119048, Russia; artemmozherov@gmail.com

**Keywords:** decellularized, tissue-engineered, matrix, islet of Langerhans

## Abstract

The treatment of type 1 diabetes through pancreatic islet transplantation faces significant limitations, including donor organ shortages and poor islet survival due to post-transplantation loss of extracellular matrix support and inadequate vascularization. Developing biocompatible scaffolds that mimic the native islet microenvironment could substantially improve transplantation outcomes. This study aimed to create and evaluate decellularized (DCL) matrices from porcine organs as potential platforms for islet transplantation. Porcine lung and pancreatic tissues were decellularized using four different protocols combining detergents (Triton X-100, SDS and SDC) with optimized incubation times. The resulting matrices were characterized through DNA quantification and histological staining (H&E and Van Gieson). Islet viability was assessed in vitro using Live/Dead staining after 3 and 7 days of culture on the matrices. In vivo biocompatibility was evaluated by implanting matrices into rat omentum or peritoneum, with histological analysis at 1-, 4-, and 8 weeks post-transplantation. Protocols 3 (for lung tissue) and 4 (for pancreas tissue) demonstrated optimal decellularization efficiency with residual DNA levels below 8%, while preserving the collagen and elastin networks. In vitro, islets cultured on decellularized lung matrix had maintained 95% viability by day 7, significantly higher than the controls (60%) and pancreatic matrix (83%). The omentum showed superior performance as an implantation site, exhibiting minimal inflammation and fibrosis compared to the peritoneum sites throughout the 8-week study period. These findings establish DCL as a promising scaffold for islet transplantation due to its superior preservation of ECM components and excellent support of islet viability. This work provides a significant step toward developing effective tissue-engineered therapies for diabetes treatment.

## 1. Introduction

Type 1 diabetes mellitus (DM1) is a chronic condition characterized by the autoimmune destruction of pancreatic beta cells, leading to absolute insulin deficiency and persistent hyperglycemia [1]. While conventional insulin therapy remains the cornerstone of treatment, it fails to fully replicate the native physiological regulation of blood glucose levels, often resulting in long-term complications [2]. Pancreatic islet transplantation has emerged as a promising therapeutic alternative, offering the potential to restore endogenous insulin production and achieve glycemic control [3]. Despite its clinical benefits, this approach faces significant challenges, including a critical shortage of donor pancreata and the limited survival of transplanted islets due to the loss of extracellular matrix (ECM) support and impaired vascularization [4]. The ECM plays a pivotal role in maintaining islet function by providing structural integrity, facilitating cell–matrix interactions, and promoting angiogenesis [5]. During isolation, islets are stripped of their native ECM, which exacerbates their vulnerability to hypoxia and inflammatory responses post-transplantation. To address these limitations, tissue-engineering approaches have been explored to create scaffolds that mimic the native pancreatic microenvironment, thereby enhancing islet survival and functionality. Among these, DCL matrices derived from native tissues have gained prominence due to their preserved ECM architecture, biocompatibility, and ability to support cell adhesion and proliferation [6]. These matrices retain critical ECM components such as collagen, elastin, and adhesive proteins, which are essential for recellularization and subsequent graft integration [7]. However, the scarcity of human donor organs has necessitated the exploration of xenogeneic sources, with porcine tissues being viable candidates due to their anatomical and physiological similarities to human tissues. Recent studies have demonstrated the potential of DCL matrices derived from organs such as the lung, which offers a highly vascularized and elastic ECM, as an alternative scaffold for islet transplantation. Nevertheless, the success of such approaches hinges on the optimization of decellularization protocols to ensure sufficiently complete cellular removal while minimizing ECM damage. The primary objective of this study was therefore to evaluate the efficiency of existing decellularization protocols for porcine lung and pancreatic matrices and to optimize these protocols for subsequent recellularization with pancreatic islets. Additionally, we investigated the choice of transplantation site as this is critical since it can influence both the inflammatory response and the vascularization of the graft. Our findings contribute to the growing body of research focused on overcoming the limitations of current islet transplantation methods and advancing cell therapy approaches for the treatment of diabetes.

## 2. Results

### 2.1. Evaluation and Optimization of Decellularization Protocols for Lung and Pancreatic Tissues: Residual DNA Analysis and ECM Integrity Assessment

The efficacies of the decellularization protocols were assessed through quantitative residual DNA analysis and histological evaluation of extracellular matrix (ECM) integrity. The initial protocols (1 and 2), derived from the literature, as applied to both lung and pancreatic tissues, were tested for their ability to remove cellular components while preserving ECM architecture.

Protocol 1 demonstrated poor structural preservation for both organs. Histological examination using hematoxylin–eosin (H&E) and Van Gieson (VG) staining revealed fragmented collagen networks in the lung matrices, including disrupted alveolar septa and irregular elastin distribution. Similarly, pancreatic matrices processed using Protocol 1 exhibited collapsed interlobular connective tissue septa and disorganized vascular collagen frameworks. Residual DNA levels exceeded acceptable thresholds in both organs (lung: >10%; pancreas: >8%, correlating with incomplete cellular removal and ECM damage (Table 1).

Following Protocol 2, residual DNA levels remained evident >8%), indicating incomplete decellularization. Additionally, pancreatic matrices processed using Protocol 2 exhibited severe structural degradation, including fragmented interlobular septa and disruption of their dense collagen networks. In contrast, the lung matrices retained near-native architecture under Protocol 2, with intact alveolar septa and preserved perivascular collagen bundles. This disparity underscores the lung’s relative resistance to moderate detergent exposure, whereas pancreatic tissue—originally characterized by a denser extracellular matrix—required more stringent protocol optimization to retain this.

To address these limitations the protocols were individually optimized as Protocols 3 (for lung) and Protocol 4 (for pancreas). Analysis of the results of these treatments revealed a statistically significant reduction in DNA content following decellularization, with residual DNA levels decreasing to 0.46 ± 0.2 μg/mg for DCL lung (*p* < 0.05) and 0.36 ±  0.1 μg/mg for DCL pancreas (*p* < 0.05) compared to native tissues, confirming effective cellular component removal (Figure 1).

Histological examination using hematoxylin and eosin (H&E) staining demonstrated alveolar architecture of the lung and acinar architecture of the pancreas without pathological changes (Figure 2). No cells or residual cellular detritus were detected in the matrices after the decellularization procedure (Figure 2B,D blue and yellow arrows). Van Gieson (VG) staining demonstrates the structure of collagen fibers of the ECM, which take on a bright pink color. Collagen fibers form the base of the alveolar septa (pink arrows), vascular and bronchial walls (green arrows) of the lung, and are preserved in the DCL matrix (Figure 2A,B). In the pancreas, collagen fibers make up thin interacinar septa and thicker interlobular septa (orange arrows), which are also preserved in the DCL matrix (Figure 2C,D). Preservation of the native collagen structure during histological examination confirms the stability and structural integrity of the DCL matrices.

The combination of biochemical and morphological data substantiates that Protocols 3 and 4 successfully balance aggressive cellular clearance with ECM preservation, thereby meeting established criteria for tissue-engineering applications where scaffold ultrastructure and biocompatibility are paramount.

The comparative analysis underscored the superiority of the adapted protocols. While Protocol 1 failed to balance DNA removal with ECM conservation, Protocol 2 did partially preserve lung structure; however, its failure to maintain pancreatic integrity and to reduce DNA to acceptable levels rendered it unsuitable for clinical application. Conversely, Protocols 3 and 4 balanced effective decellularization with ECM preservation, establishing their utility as scaffolds for tissue-engineered constructs.

### 2.2. Isolation of Pancreatic Islets and Recellularization of DCL Matrices

The cytotoxicity of the obtained matrices was assessed in vitro on an islet of Langerhans cells isolated from rats.

Cells cultured without matrix were used as controls. Cells cultured on decellularized lung and pancreas matrices served as the experimental test materials (Figure 3).

On the 3rd and 7th days of maintaining both matrix-based and matrix-free islets in culture, cell viability was assessed by staining using a Live/Dead Cell Double Staining Kit (Sigma-Aldrich, St. Louis, MI, USA). Based on the calculation of the ratio of living and dead islets on the matrices, as well as those of islets without a matrix, using the ImageJ 1.43 u software program, it was found that the percentage of living islets of Langerhans cells without a matrix on the 7th day of incubation had statistically significantly decreased—60% rather than 82% (Figure 4).

However, it was demonstrated that, by the 7th day of incubation, the percentage of viable islets of Langerhans was statistically significantly greater when incubated on decellularized matrices compared to islets of Langerhans incubated without matrix: 95% on decellularized lung matrix and up to 83% on decellularized pancreatic matrix compared with 60% for islets of Langerhans without any matrix.

It should also be noted that, on the 7th day of incubation of the islets of Langerhans on the decellularized lung matrix, the viability of the cells was higher than on the decellularized pancreatic matrix. This is perhaps associated with the presence of a larger number of elastin fibers in the decellularized lung matrix.

Based on an analysis of the data obtained, a tendency towards a larger decrease in cell viability for incubation not on matrices from the 3rd to the 7th days, as well as a greater retention of cell viability during incubation on decellularized matrices created from both the lung and pancreas for the corresponding period, was noted, indicating the absence of cytotoxicity of the obtained matrices. It should also be noted that, on the 7th day of incubation of the islets of Langerhans, the percentage viability on the DCL of the lung was higher than that for incubation on the DCL of the pancreas. We postulate that this is associated with the presence of elastin fibers in the DCL of the lung. Based on the results of our assessment of the cytotoxicity of the matrices in vitro, it can be concluded that the lung can serve as a potentially better alternative organ for the creation of tissue engineering constructs (TICs) for use in the pancreas.

### 2.3. In Vivo Evaluation of Matrix Biodegradation and Biocompatibility

The in vivo biodegradation and biocompatibilities of DCL lung and DCL pancreas matrices were evaluated following transplantation into the omentum and peritoneum of Wistar rats. Histological examination at 1, 4, and 8 weeks after transplantation revealed matrices degradation, as well as inflammatory and fibrotic reactions, dependent on the duration and site of implantation, but not on the matrix type.

At 1 week, degradation of the matrix structure with partial fragmentation and material engulfment by macrophages was observed both in the omentum and in the peritoneum. An acute productive inflammatory reaction developed around the matrices, which included cellular infiltration of histiocytes and lymphocytes, and single neutrophils (Figure 5A,D). The density of the inflammatory infiltrate did not differ significantly between the peritoneum and omentum, or between the lung DCL and the pancreatic DCL (Figure 6). At 4 weeks, the matrix material was fragmented particles lying among the inflammatory infiltrate and newly formed connective tissue; foreign body giant cells and macrophages that had engulfed the matrix material were observed (Figure 5B,E). The density of the inflammatory infiltrate was significantly higher in the samples implanted into the peritoneum (Figure 6). Connective tissue proliferation was also more pronounced in the peritoneum (Figure 5B,E). At 8 weeks, the cellular infiltrate had decreased, although fibrosis remained evident, particularly in the peritoneal grafts (Figure 5C,F).

Notably, the omentum demonstrated a milder fibrotic reaction and reduced inflammatory infiltration compared to the peritoneum. No adverse systemic reactions or signs of acute rejection were observed, confirming the biosafety of the xenogeneic matrices. These results suggest that the omentum is a more favorable transplantation site due to its enhanced biocompatibility and reduced fibrotic encapsulation, supporting its potential use in future clinical applications for pancreatic islet transplantation. However, the persistence of fibrosis around the matrices, even at later time points, highlights the need for further optimization of such decellularization protocols to minimize host immune responses and to improve long-term graft survival. We did additionally count the density of lymphocyte infiltration in the area around the grafts, which allowed us to assess the overall level of inflammatory reaction in the tissue (Figure 6).

## 3. Discussion

In this study, we successfully optimized decellularization protocols for porcine lung and pancreatic tissues, achieving effective cellular removal while preserving ECM integrity.

The choice of porcine lung and pancreatic tissues for decellularization in this study was driven by their unique structural and functional properties, which make them promising candidates for creating scaffolds to support pancreatic islet transplantation in humans. The pancreas was prioritized as the principal source for decellularized matrices since its native ECM represents the islets’ natural microenvironment, having originally housed these endocrine cell clusters in vivo. A pancreas-derived matrix thus offers an organ-specific scaffold that recapitulates the original niche conditions required for proper islet physiology. However, the lung emerged as a viable substitute because its highly vascularized and elastic ECM architecture facilitates efficient oxygenation and nutrient diffusion—critical factors for maintaining transplanted islet viability [8]. This rationale aligns with previous studies demonstrating the lung’s potential as a scaffold source for tissue engineering applications [9]. By capitalizing on these benefits, our study sought to develop optimized platforms for islet transplantation.

We tested various decellularization protocols based on the literature (Protocol 1 and Protocol 2) [10,11]. We chose these protocols because of their application of the detergents most often used to obtain decellularized tissues. However, initial Protocols 1 and 2 demonstrated significant limitations, failing to achieve the required delicate balance between effective cellular removal and extracellular matrix preservation. An optimal decellularization protocol must maintain the ECM’s ultrastructure—including its collagen and elastin fiber networks with associated adhesive proteins for proper scaffold functionality, while simultaneously minimizing residual cellular DNA content below biologically acceptable thresholds to ensure biocompatibility. Since the above protocols did not show the required preservation of matrix structure and their decellularization activity did not sufficiently reduce the amount of residual DNA, a modification of Protocol 2 was tested with the lung and pancreas source tissues.

Sodium dodecyl sulfate (SDS), sodium deoxycholate (SDC), and Triton X-100 are among the most widely used ionic decellularization agents due to their efficiency in solubilizing cytoplasmic membranes, lipids, and DNA [12]. It is very important to select appropriate detergent concentrations to preserve the matrix structure of the native organ while minimizing residual DNA. For the lung, increasing the soak time in a low SDS concentration was suggested to achieve greatest DNA minimization. For the pancreas, using only SDS of varying concentrations as a chemical detergent was proposed since SDC and Triton X-100 destroy the structure of the pancreatic matrix.

The in vitro evaluation revealed that the decellularized lung (DCL lung) and pancreatic (DCL pancreas) matrices significantly enhanced the viability of rat pancreatic islets compared to the matrix-free controls. This difference may be attributed to the better cellular adherence to the lung matrix’s rich elastin content and porous architecture, likely contributing to enhanced nutrient diffusion and mechanical support, factors known to promote islet function. These results corroborate the observations of authors who attempted to create TICs based on a decellularized lung [13]. They incubated cells both on and off the matrix, while measuring their insulin secretion. It has been shown that, on DCL lung, cells can remain viable for 5 days. The viability of islets of Langerhans is slightly higher when incubated on a pulmonary matrix compared with their incubation without a matrix. The authors created TICs based on a decellularized pancreas. They found it was possible to incubate mesenchymal stem cells (MSCs) for 3 days on the obtained pancreatic matrix [14]. While it has also been experimentally proven that incubation on a matrix increases the viability of MSCs, it should be noted that incubation of islets of Langerhans may have its own characteristics, namely, cell viability may not change for 3 days. The authors attempted to create a TICs based on a hydrogel and a floating island-shaped culture [15]. On both the 7th and 14th days of co-cultivation, the preservation of a significant number of hormonally active beta cells was confirmed. This indicated the absence of any negative effect of the three-dimensional biomatrix on the survival and morphofunctional qualities of the cells. At the same time, the attachment of such cultures to the surface of the biomatrix was noted. The authors observed incubation of Langerhans islets on a decellularized pericardial matrix using fluorescent staining of living and dead cells [16]. The results showed that all the islets remained viable after 24 h. On the contrary, islets cultivated without matrix showed a tendency to aggregation, which led to the formation of necrotic areas.

In addition, we have identified the omentum as a preferred transplantation site due to its reduced inflammatory response and fibrotic encapsulation compared to the peritoneum. Choosing the site for transplantation is a non-trivial task. Currently, the standard approach involves transplanting islets into the portal vein of the liver [17]. However, acute inflammatory reactions mediated by blood (instant blood-mediated inflammatory reaction, IBMIR) can lead to massive graft loss at the earliest stages. Moreover, the portal vein of the liver is not suitable for the transplantation of islets within matrices because of their size. A limited number of studies have been aimed at in vivo testing of decellularized matrices for the creation of pancreatic TICs. The authors transplanted a decellularized fascia sample into the rabbit peritoneum with a biopsy of this tissue being performed 15 days later [18]. The graft was incorporated into a full-layered defect made in the rabbit’s abdominal wall. This tissue was infiltrated by granulation and inflammatory cells, the histological structure being preserved 15 days after surgery. Karimi S et al., 2022 conducted in vivo transplantation of testis–extracellular matrix (testis–ECM) scaffolds and demonstrated appropriate positions for transplantation allowing angiogenesis but low infiltration of inflammatory cells [19]. The evaluations in our work highlighted the omentum’s advantages as a transplantation site, with both minimal inflammation and fibrosis at 8 weeks post-implantation when compared to the peritoneum. This highly vascularized tissue secretes various growth factors (e.g., CXCR4, VEGF, and SDF-1) that promote vascularization and survival of the islets [17]. While decellularization effectively reduces immunogenicity by removing cellular components, the host response to implanted matrices remains a natural part of the engraftment process [20]. Like many biomaterials, decellularized matrices undergo controlled biodegradation, during which latent peptides and growth factors are gradually released, contributing to tissue remodeling [21]. Histological evaluations typically reveal transient inflammatory infiltration, macrophage activity, and the presence of multinucleated giant cells—all indicative of a foreign body response [14,21].

For instance, MRI tracking of decellularized pancreatic matrices in rat omentum showed a sixfold volume reduction within the first two weeks post-transplantation, reflecting the expected loss of the native organ’s bulk structure [14]. Our histological findings support this biphasic process: early macrophage-mediated restructuring is followed by progressive ECM remodeling, consistent with the typical healing trajectory for biological scaffolds [21].

These observations underscore that mild inflammation and gradual biodegradation are inherent to the host–scaffold interaction and do not preclude successful graft integration. Instead, they reflect the dynamic balance between matrix resorption and tissue regeneration—a feature shared by many clinically approved biomaterials. Future refinements in decellularization protocols may further modulate these responses, but the current results affirm that such reactions are both anticipated and manageable within the context of functional islet transplantation.

These findings suggest that the omentum should be strongly considered as a transplantation site.

While the omentum demonstrated superior biocompatibility compared to the peritoneum, the observed inflammatory response underscores the potential need for immunosuppressive strategies in clinical applications. Immunosuppression is particularly critical for xenogeneic matrices, as residual cellular components or ECM epitopes may trigger immune recognition [7]. Systemic immunosuppressants like tacrolimus or cyclosporine could mitigate this response but carry risks of toxicity and increased susceptibility to infections [17]. Alternatively, localized delivery of immunomodulatory agents—such as anti-inflammatory cytokines or small-molecule inhibitors—to the transplant site may offer targeted control of inflammation while minimizing systemic side effects [22]. Another promising approach involves recellularized matrices with mesenchymal stem cells (MSCs), which can modulate the immune microenvironment by promoting anti-inflammatory macrophage polarization (M2 phenotype) and regulatory T-cell recruitment, thereby enhancing graft survival [14]. However, the efficacy of such strategies depends on the degree of decellularization, as residual DNA or cellular debris may exacerbate immunogenicity [12]. Future studies should explore combinatorial therapies, such as MSC-seeded matrices coupled with localized immunosuppression, to optimize outcomes. Our findings highlight the importance of tailoring immunosuppressive protocols to both the scaffold properties and the transplantation site, with the omentum’s inherent immunoregulatory capacity offering a distinct advantage for clinical transplantation.

While decellularized matrices have demonstrated significant promise for islet transplantation, alternative strategies such as encapsulating pancreatic islets within hydrogels have also been explored [23]. Hydrogel-based encapsulation aims to immunoprotect islets while permitting nutrient exchange, yet clinical translation remains hindered by several limitations. A key advantage of decellularized matrices lies in their capacity to facilitate full vascular integration. Unlike encapsulated islets, which are confined by semipermeable membranes, matrix-embedded islets directly interact with host vasculature, enhancing oxygen and nutrient diffusion critical for long-term viability [8]. Encapsulation systems often compromise islet function due to hypoxic stress and delayed insulin kinetics, as hormonal secretion must traverse the hydrogel barrier [23]. Moreover, ultrathin encapsulation membranes may permit immune cell infiltration, while conformal coatings risk islet damage through mechanical stress or penetration into cellular structures [24]. Retrieval of nanoencapsulated grafts also poses technical challenges, potentially triggering inflammatory responses.

Hydrogels designed to mimic native ECM components, such as collagen-based scaffolds, can improve peri-graft vascularization [24]. However, their efficacy depends on stringent biocompatibility standards, an area where decellularized matrices excel due to their inherent ECM composition. Matrix-based transplantation eliminates diffusion barriers, enabling rapid glucose sensing and insulin release, thereby optimizing glycemic control. Furthermore, decellularized scaffolds preserve islet morphology by replicating native biomechanical and biochemical cues, reducing apoptosis and endoplasmic reticulum stress [5]. Immunologically, carefully processed decellularized matrices exhibit minimal antigenicity, whereas capsules frequently induce fibrotic encapsulation, impairing graft function over time [20].

Decellularized matrices also offer unparalleled versatility for bioengineering. Their ECM architecture can be functionalized with angiogenic factors (e.g., VEGF), immunomodulators, or cytokines to tailor the graft microenvironment [22]. This level of customization is unattainable with standardized encapsulation systems. Recent advances combining decellularized scaffolds with genetically modified islets—such as HLA-silenced or CD47-overexpressing variants—further enhance immune evasion and graft survival [24]. These hybrid approaches synergize the structural benefits of ECM scaffolds with cutting-edge gene-editing technologies, representing a paradigm shift in diabetes therapy.

In summary, while encapsulation strategies provide partial solutions, decellularized matrices surpass them in vascular integration, functional performance, and immunological compatibility. Future research should focus on combinatorial therapies, such as endothelial cell-seeded matrices or localized immunosuppression, to address residual challenges in graft revascularization and immune rejection. Such innovations will be pivotal in advancing tissue-engineered therapies toward clinical viability for type 1 diabetes.

## 4. Materials and Methods

### 4.1. Experimental Design

This study aimed to evaluate decellularized lung (DCL lung) and pancreatic (DCL pancreas) matrices as scaffolds for pancreatic islet transplantation. Porcine lung and pancreatic tissue samples were collected and processed for decellularization (Figure 7). Tissue fragments (not whole organs) were decellularized using two standard protocols (Protocols 1 and 2), which were applied to both lung and pancreas. These protocols were subsequently modified to optimize decellularization efficiency, resulting in Protocol 3 for lung tissue and Protocol 4 for pancreatic tissue. The efficacy of the protocols was evaluated based on histological analysis and residual DNA quantification.

The decellularized matrices produced by the optimal protocols (3 for lung and 4 for pancreas) were further tested. For in vitro assessment, the matrices were recellularized with pancreatic islets, and islet viability was evaluated. For in vivo evaluation, the matrices were transplanted into two sites in rats—the omentum and the peritoneum—to assess biocompatibility, inflammatory response, and fibrosis over time.

### 4.2. Animals and Sample Collection

All experimental procedures involving animals were approved by the Institutional Animal Ethics Committee of Privolzhsky Research Medical University (Protocol No. 10; Date: 26 June 2020). This study utilized Wiesenau minipigs (3–4 months old, *n* = 9) and Wistar rats (1 month old, *n* = 30). The minipigs served as donors for the preparation of decellularized pancreas and lung scaffolds, while the rats were employed for pancreatic islet isolation and as DCL transplantation recipients. The rats were pair-housed under standardized environmental conditions with a 12 h light–dark cycle and acclimated to routine handling and restraint. The minipigs were individually maintained at 19–23 °C under a 12:12 h day/night cycle. Prior to euthanasia, the rats were anesthetized via intramuscular injection of Zoletil (Virbac NZ, Hamilton, New Zealand; 6 mg/kg) and xylazine (Bayer, Leverkusen, Germany; 90 mg/kg). The minipigs received premedication with Zoletil (10 mg/kg) and xylazine (4 mg/kg), followed by continuous intravenous infusion of Propofol–Lipuro (Braun, Melsungen, Germany; 16 mg/kg/h). Post-euthanasia, the minipig organs were immersed in primary decellularizing medium to generate acellular matrices. The rat pancreata were collected into 0.5% bovine serum albumin (BSA) solution for subsequent islet isolation. Following DCL transplantation, the transplant sites were excised and fixed in 10% neutral-buffered formalin for histological examination.

### 4.3. Selection of Organs and Decellularization Protocols

The pancreas was chosen as the primary source for decellularized matrices due to its native ECM, which naturally supports islet survival and function. The lung was selected as a potential alternative because its highly vascularized and elastic ECM structure promotes the oxygenation and vascularization critical for transplanted islet viability [8].

The selection and optimization of decellularization protocols in this study were conducted through a systematic evaluation of existing methods, followed by targeted modifications to address identified limitations. The initial protocols (Protocols 1 and 2) were chosen based on the established literature for decellularizing both lung and pancreatic tissues [10,11]. However, experimental testing revealed that these methods were insufficient to provide complete removal of the cellular components while preserving the critical ECM architecture. To overcome these shortcomings, two organ-specific protocols were developed. Protocol 3 was optimized exclusively for lung tissue by extending the incubation period in SDC to 41 h. For pancreatic tissue, Protocol 4 was designed with a refined SDS-based approach, incorporating a graded series of SDS concentrations (0.1% in distilled water, 0.1% in 1 N NaCl, and 0.1% in PBS) with carefully timed incubations. For the lung tissue, the increased soak time in low SDS concentration was intended to achieve as much DNA minimization as possible. For the pancreas, using only SDS of varying concentrations as a chemical detergent was suggested, since the SDC and Triton X-100, as used in Protocol 2, destroy the structure of the pancreatic matrix.

Protocol 1 was applied to both lung and pancreatic tissues and involved sequential immersion in detergents: samples were first treated with 0.5% Triton X-100 for 1 h, followed by 0.5% SDS for 1 h, and then 1% SDC for 1 h. Finally, the tissues were incubated in 0.075% SDC under continuous shaking for 24 h.

Protocol 2, also tested on both organs, followed a similar detergent sequence (Triton X-100 → SDS → SDC) but extended the final SDC incubation to 41 h for lung tissue, while the pancreatic samples were processed with 0.1% SDS in distilled water (1 h), 0.1% SDS in 1 N NaCl (1 h), and 0.1% SDS in PBS (18 h).

Protocol 3 was specifically optimized for lung decellularization by further prolonging the SDC incubation to 41 h after the initial Triton X-100 and SDS steps, ensuring thorough cell removal while preserving ECM integrity.

Protocol 4, tailored for the pancreas, utilized a refined SDS-based approach: tissues were treated with 0.1% SDS in distilled water (1 h), 0.1% SDS in 1 N NaCl (1 h), and 0.1% SDS in PBS (18 h).

All protocols included antibiotic–antimycotic supplementation and were conducted under sterile conditions. Post-decellularization, the matrices were washed in Hanks’ solution (5 days) and DMEM (7 days) to remove residual detergents and enhance biocompatibility.

### 4.4. In Vitro Evaluation of DCL Matrices

Residual DNA content was assessed to evaluate the efficiency of decellularization. Tissue samples (native and decellularized) were enzymatically digested using 1% collagenase and trypsin, followed by DNA extraction with the *ExtractDNA Blood & Cells* kit (Evrogen, Moscow, Russia). DNA concentration was measured spectrophotometrically (*NanoDrop One*, Thermo Scientific, Waltham, MA, USA) at 260/280 nm.

For histological examination, the native and decellularized tissue samples were fixed in 10% neutral buffered formalin for 48 h to ensure proper preservation of the tissue architecture. Following fixation, the samples were processed through a standard dehydration series using graded ethanol solutions, then cleared in xylene, and embedded in paraffin blocks. Tissue sections of 7 μm thickness were cut using an HM 325 microtome (Thermo Scientific, USA) and mounted on glass slides. The sections were then subjected to two main staining methods: hematoxylin and eosin for general histological assessment of tissue morphology and Van Gieson staining for specific visualization of collagen fibers (stained bright pink) in the extracellular matrix. The stained sections were examined under an EVOS m7000 imaging system in transmitted light (Invitrogen, Thermo Fisher Scientific, Waltham, MA, USA).

### 4.5. Isolation of Pancreatic Islets and Recellularization of DCL Matrices

Pancreatic islets were isolated from Wistar rats through enzymatic digestion using collagenase V (Sigma, St. Louis, MI, USA) in modified Hanks’ solution supplemented with CaCl_2_. Following perfusion, the pancreas was incubated at 37 °C for 11–15 min with gentle agitation until tissue dissociation was complete. The digested material was filtered through a 0.5 mm mesh sieve to remove undigested fragments, and the islets were purified by density gradient centrifugation using Ficoll DL-400 (1.048–1.095 g/mL) (Sigma, Saint Louis, MI, USA). Islet purity and viability were confirmed by dithizone staining, with subsequent culture in RPMI-1640 medium (Gibco, London, UK) containing 10% FBS, L-glutamine, and antibiotics at 37 °C under 5% CO_2_.

For recellularization, the decellularized lung and pancreatic matrices (5 μm slices) were seeded with 3500 islets/well in DMEM/RPMI (1:4) medium. Islet viability was evaluated on days 3 and 7 using the Live/Dead Cell Double Staining Kit (Sigma-Aldrich, St. Louis, MI, USA), where calcein-AM (20 μg/mL) labeled any live cells (green) and propidium iodide (10 μg/mL) was used to identify dead cells (red). The resulting fluorescence images were quantified with ImageJ to determine the viability percentages. Control islets, for comparative analysis, were maintained under identical conditions but without the matrices.

### 4.6. In Vivo Evaluation of Matrix Biodegradation and Biocompatibility

To assess biodegradation and biocompatibility, decellularized lung and pancreatic matrices (5 μm slices) were transplanted into two sites in the Wistar rats: the omentum (selected for its high vascularization and growth factor secretion [CXCR4, VEGF and SDF-1]) and the peritoneum (a standard transplantation site) [17]. Prior to surgical implantation, the animals were anesthetized with Zoletil (6 mg/kg) and xylazine (90 mg/kg). The matrices were positioned in both sites through 0.5–0.7 cm abdominal incisions, followed by layered closure (Figure 8).

For omental implantation, a matrix slice was carefully placed within the omentum, which was then fashioned into a pouch to securely encapsulate the graft. The pouch was closed with multiple sutures to ensure stable retention of the matrix. Following this procedure, the omentum was returned to the abdominal cavity, and the peritoneum and skin were sutured in layers. Notably, the matrix itself was not directly sutured to either the omentum or peritoneum, allowing natural integration.

Postoperative care included antibiotic prophylaxis (Augmentin, 1 mg/mL at 3–5 days). At 1-, 4-, or 8 weeks post-implantation, the graft sites were harvested and evaluated histologically. The extents of any fibrosis and inflammatory responses were assessed in each peri-graft region (the recipient’s tissue surrounding the graft) using H&E staining, with emphasis on leukocyte infiltration and the state of the ECM. Lymphocyte density in the peri-graft region was assessed in the field of view of a 40× objective (0.196 mm^2^), in 5 fields of view for each sample.

### 4.7. Statistical Analysis of Data

Three independent experiments and three technical replicates were conducted for each of the protocols: for effectiveness of decellularization, cytotoxicity, and biosafety of the resulting matrices. Statistical data processing was performed using R-studio version R 3.6.0+ (RStudio PBC, Boston, MA, USA) and MS Excel version 2107 (Microsoft, Redmond, WA, USA). Initially, the data distribution was checked using the Shapiro–Wilk criterion. Student’s *t*-test, with Bonferroni correction for multiple comparisons, served as a criterion for the comparing groups. The data were presented as the mean with standard deviations (mean ± SD). The results were considered statistically significant at *p* < 0.05.

## 5. Conclusions

This study optimized decellularization protocols for porcine lung and pancreatic tissues, achieving effective cell removal while preserving ECM integrity. The refined protocols (3 and 4) reduced residual DNA (<8%) and maintained key structural components. In vitro DCL lung and DCL pancreatic matrices enhanced islet viability. In vivo testing demonstrated the omentum to be an excellent transplantation site, exhibiting significantly better biocompatibility with reduced inflammatory response and minimal fibrosis compared to peritoneal implantation.

## Figures and Tables

**Figure 1 ijms-26-06692-f001:**
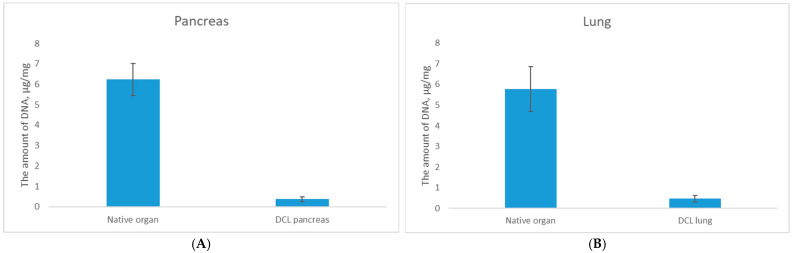
Residual DNA analysis of (**A**) native organ and DCL pancreas; (**B**) native organ and DCL lung.

**Figure 2 ijms-26-06692-f002:**
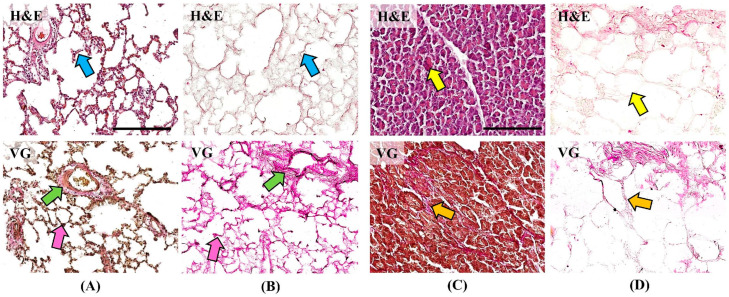
Histological study of native organs (**A**,**C**) and DCL matrices (**B**,**D**) prepared according to protocols 3 and 4. (**A**) Native alveolar lung structure—blue arrow; interalveolar septum with cellular elements—green arrow; collagen fibers in the vascular wall—pink arrow; collagen fibers of the interalveolar septum. (**B**) DCL lung—blue arrow; alveolar septum without cells—green arrow; dense accumulation of collagen fibers at the site of the vascular wall—pink arrow; collagen fibers of the thin alveolar septum. (**C**) Native acinar pancreas structure—yellow arrow; acinar cells—orange arrow; interacinar septa. (**D**) DCL pancreas—yellow arrow; cells are absent—orange arrow; intact collagen fibers of the septa (scale bar 200 μm).

**Figure 3 ijms-26-06692-f003:**
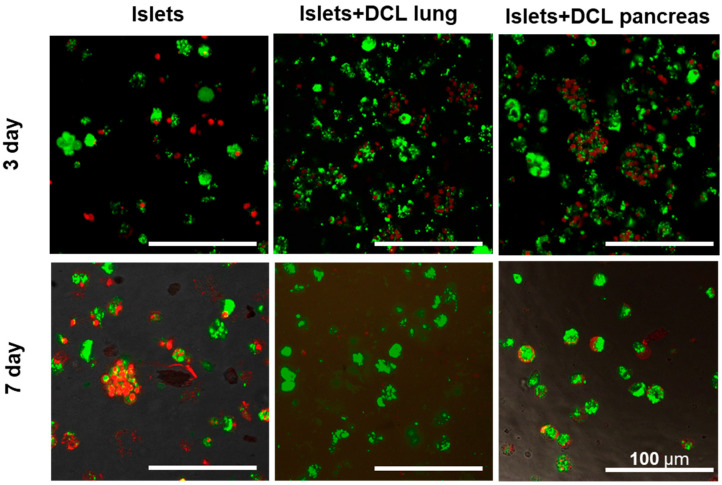
Fluorescence microscopy of islets of Langerhans, islets on DCL lung, islets on DCL pancreas (left to the right) on 3 and 7 days of culture Live/Dead staining Calcein AM (green) and propidium iodide (red) (scale bar 100 μm).

**Figure 4 ijms-26-06692-f004:**
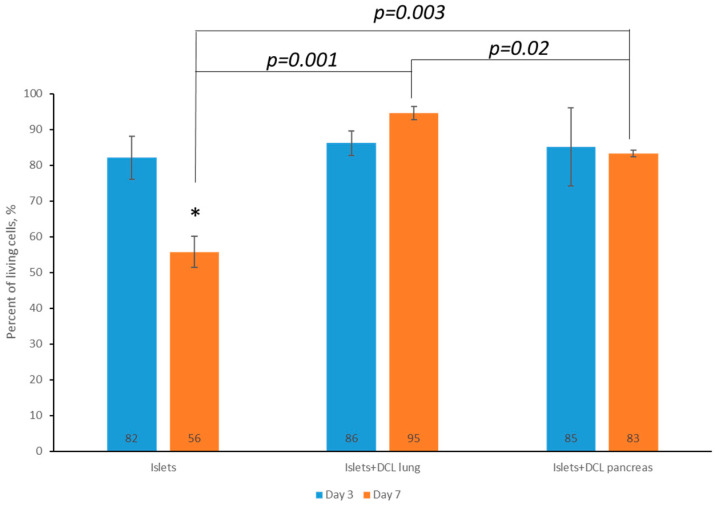
Percentage of living islets of Langerhans without matrices on DCL lung and DCL pancreas on the 3rd and 7th days, (Mean ± SD, *—*p* < 0.05).

**Figure 5 ijms-26-06692-f005:**
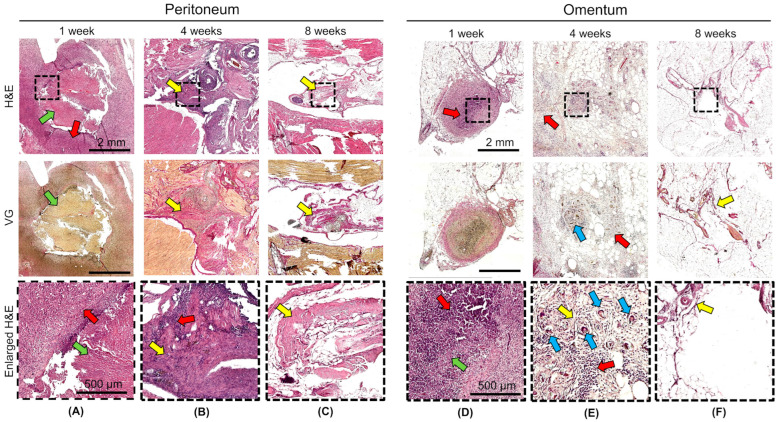
Histological examination of the DCL lung and DCL pancreas matrices during transplantation into the peritoneum (**A**–**C**) and omentum (**D**–**F**) at 1, 4, and 8 weeks. (**A**) Acute productive inflammatory reaction to the graft in the peritoneum after 1 week—green arrow; graft material—red arrow; dense inflammatory cell infiltrate—no connective tissue was detected on VG staining. (**B**) Connective tissue proliferation at the graft site in the peritoneum after 4 weeks—yellow arrow; connective tissue—red arrow; foci of lymphocytic inflammatory infiltrate. (**C**) Resolution of inflammation and residual connective tissue (yellow arrow) in the peritoneum after 8 months. (**D**) Acute productive inflammatory reaction to the graft in the omentum after 1 week—red arrow; inflammatory cell infiltrate—green arrow; fragmented matrix material. (**E**) Connective tissue proliferation at the graft site in the omentum after 4 weeks—yellow arrow; loose connective tissue—red arrow; inflammatory cells—blue arrows; multinucleated foreign body giant cells. (**F**) Resolution of inflammation and isolated residual connective tissue bundles in the omentum after 8 weeks. The black dotted square on the histological images corresponds to the enlarged (H&E) areas. H&E—hematoxylin-eosin; VG—Van Gieson staining (scale bar on images 2 mm, scale bar on enlarged H&E image 250 µm).

**Figure 6 ijms-26-06692-f006:**
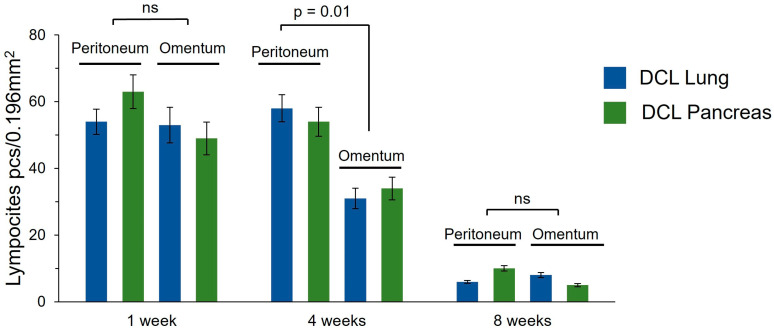
Lymphocyte density (pcs/0.196 mm^2^) in the area of matrix implantation into the peritoneum and omentum. ns—nonsignificant (*p* ≥ 0.05).

**Figure 7 ijms-26-06692-f007:**
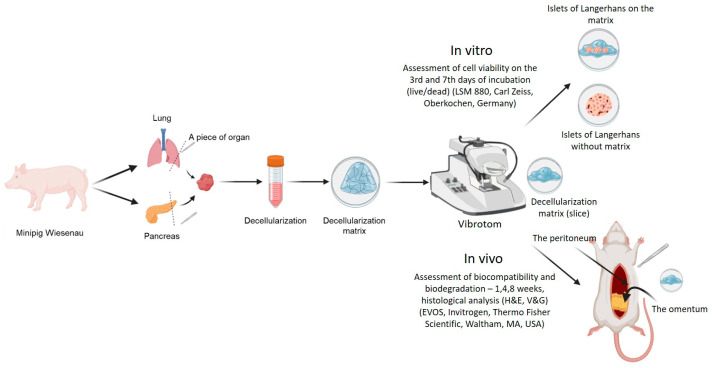
The scheme of the experiment. Created in Biorender. Alexandra Bogomolova. (2025) https://app.biorender.com (accessed on 12 June 2025).

**Figure 8 ijms-26-06692-f008:**
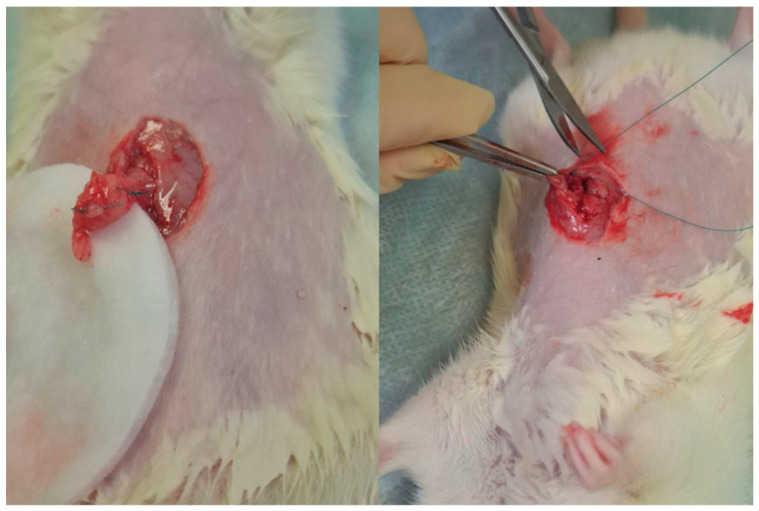
The process of forming a pouch in the omentum and a pocket in the peritoneum for matrix transplantation.

**Table 1 ijms-26-06692-t001:** Comparison and optimization of organ decellularization protocols.

Protocol Number	The Amount of DNA, %		Preservation of the Structure	
	Pancreas	Lung	Pancreas	Lung
**No. 1** Supersaturated saline solution 1.19 M KCl, 1.74 M NaCl, 0.86 M CaCl_2_-96 h	>90%	>90%	+	+
**No. 2** 0.5%Triton X-100-1 h 0.5% SDS-1 h 1% SDC-1 h 0.075% SDS-24 h	>20%	>70%	-	+
**No. 3** 0.5%Triton X-100-1 h 0.5% SDS-1 h 1% SDC-1 h 0.075% SDS-41 h		<8%		+
**No. 4** 0.1% SDS-1 h 0.1% SDS in 1 N NaCl-1 h 0.1% SDS-18 h	<6%		+	

## Data Availability

The data presented in this study are available on request from the corresponding author.

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
