# Peer review of "In Vitro and In Vivo Testing of Decellularized Lung and Pancreas Matrices as Potential Islet Platforms"

_ijms, 2025, doi:10.3390/ijms26146692_

Round 1

Reviewer 1 Report

Comments and Suggestions for Authors

To the authors • Reviewer Comments:

  • The authors present a compelling study focused on the development and evaluation of in Vitro and in Vivo testing of decellularized lung and pancreas matrices as potential platforms for islet transplantation. This research is highly relevant to the advancement of safe and effective protocols involving organ-derived scaffolds for islet engraftment. I believe the manuscript is suitable for publication, provided that the authors address several important points-particularly through the inclusion of additional illustrative images that clarify key aspects of the methodology and results.

#1. Inclusion of Morphological Imaging. It would strengthen the manuscript if the authors included representative images (e.g., 5 μm tissue sections) of the islet platforms derived from both lung and pancreatic matrices, captured both prior to and following in vivo use. These images would provide valuable insights into morphological changes, allowing for a better assessment of platform stability and integrity. Additionally, it would be informative to include photographs documenting the gradual change in color during the decellularization process of the lung and pancreatic tail tissues.

#2. Visualization of Platform Use and Transplantation. The manuscript would benefit from a figure presenting a detailed and illustrative overview of the islet transplantation procedure, including the anatomical location of the lung and pancreas tissues prior to decellularization. A visual representation of the islet engraftment platform in situ would further clarify the methodology.

#3. Extracellular Matrix (ECM) Placement. Please consider including images of the ECM scaffolds once placed on the extruded omentum, as well as after packaging within the omental tissue. It would be helpful to see whether the ECM wrapping is secured with a single suture or multiple stitches. Such details are critical for reproducibility and for evaluating the implantation approach.

  • The manuscript currently lacks essential technical information regarding the direct analysis of the morphological properties of the decellularized matrix tissues.

#4. The authors are encouraged to provide image-based data or detailed descriptions illustrating the structural behavior of the matrices before and after transplantation, as well as following each stage of the applied protocol. Particular attention should be given to the degradation profile during the protocols of the islet platforms. These aspects are critical to fully assess the suitability, stability, and functional performance of the matrices within the transplantation context.

Reviewer 2 Report

Comments and Suggestions for Authors

This study, “In Vitro and In Vivo Testing of Decellularized Lung and Pancreas Matrices as Potential Islet Platforms”, evaluated decellularized (DCL) matrices from porcine lung and pancreas as scaffolds for islet transplantation, aiming to enhance biocompatibility and viability. Optimized protocols preserved the ECM matrices while removing cells. Islets cultured on lung matrices showed superior viability (95%) compared to pancreas matrices (83%). Omentum transplantation proved more biocompatible than peritoneum, minimizing inflammation and fibrosis. These findings suggest tissue-engineered matrices could enhance diabetes treatment strategies. It is recommended that the authors optimize the discussion section and add an inflammatory analysis to improve the quality of the manuscript further. The manuscript, after a major revision, was considered for publication in the Journal of International Journal of Molecular Sciences.

  1. In this study, the main method of the study is to investigate the short-term survival after transplantation (1, 4, and 8 weeks), thus, it is suggested to add the function of blood glucose regulation and insulin secretion after transplantation. In addition to this, it is recommended that longer-term survival and functional assessments be performed.
  2. In this study, in vivo experiments showed that omental transplants were more biocompatible than peritoneum transplants, but it was still observed through the results that there was a certain inflammatory response in the omental transplantation group. It is suggested that the authors add the analysis and quantification of inflammatory mediators in both groups. In addition, the impact of immunosuppressive strategies on the graft outcome is further analyzed in the discussion.
  3. This study emphasized the advantages of decellularized matrices, but did not provide an in-depth comparison of the differences in effectiveness between them and other islet transplantation strategies (e.g., hydrogel encapsulation, stem cell transplantation, and gene-edited islets). It is suggested that other islet transplantation techniques be added for comparison and that relevant comparisons be supplemented in the discussion section to enhance the scientific validity of a more comprehensive analysis of the strategies in this study.

Reviewer 3 Report

Comments and Suggestions for Authors

The present manuscript explores using porcine lung and pancreas decellularized extracellular matrix as potential scaffolds for improving islet viability. Overall, the concept is interesting, and the writing is generally good. However, some major concerns need to be addressed. Here are my comments:

  1. As both the lungs and the pancreas are complex organs, it is concerning how soaking them in SDS would result in complete decellularization. These organs have plenty of capillaries and small blood vessels. For complete decellularization of the whole organ, researchers usually rely on a perfusion bioreactor technique. Authors should provide adequate reasoning behind their methods.
  2. The DNA amount should be presented in ng/mg units rather than percentage form. Based on my knowledge DNA amount should be less than 50 ng per mg of tissue samples to be addressed as a completely decellularized organ. 8% or 6% of DNA is still a big number, I guess.
  3. It would be great if the authors could provide Masson’s trichrome staining of their decellularized samples, just to show how much collagen was retained after decell.
  4. Islet Equivalents (IEQ) calculation should be provided, which involves quantifying the number of islets in a preparation, considering their size and volume, and normalizing them to a standard islet size (typically 150 μm in diameter). This allows for consistent comparison of different islet preparations, regardless of their overall size distribution.
  5. Why is the total number of islets (both live and dead) decreased after 7 days (Fig. 3)? The live and dead staining figure suggests that after 7 days, islet number was decreased compared to after 3 days. I would suggest providing the total number of live and dead cells in each sample for transparency.
  6. The viability data presented in Fig. 4 are also very misleading. For example, Fig. 3 suggests that after 7 days, the number of dead cells and live cells is almost equal, but Fig. 4 shows the viability is around 83%. These discrepancies need to be resolved so that the readers don’t get confused.

Round 2

Reviewer 2 Report

Comments and Suggestions for Authors

The authors have provided appropriate revisions in response to the previous comments. The manuscript was suitable for publication.

Author Response

Review Report (Reviewer 2)

General comment: The authors have provided appropriate revisions in response to the previous comments. The manuscript was suitable for publication.
Reply: Thank you very much for your comments. We have taken your suggestions into account and made the necessary changes in the new version of the manuscript.

Reviewer 3 Report

Comments and Suggestions for Authors

The authors tried to resolve all the issues that were raised during the first revision. I have carefully reviewed the replies. However, there are still major concerns. 

  1. How did the authors calculate the viability? I presume by counting the live and dead cells, since they choose to perform a fluorescence image-based assay, counting from the images is the only viable option. If they don't want to provide the exact number, then MTT or WST assays should be done for clarity, which are more accurate for measuring viability.
  2. Still, Fig. 3 and 4 do not complement each other. After 7 days, the viability shows 83% in islet+DCL group. However, the image shows no dead cells at all! On the other hand, after 3 days, the same group shows many dead cells, but the viability is higher (85%) than after 7 days! How is it possible? These discrepancies would confuse the readers.

Round 3

Reviewer 3 Report

Comments and Suggestions for Authors

The authors resolved all the issues that were raised. I would recommend this article to be published in this journal.